# Deep Generative Neural Embeddings for High Dimensional Data Visualization

## Abstract

We propose an embedding-based visualization method along with a data generation model. In particular, corresponding locations of data points in the visualization are optimized as embeddings along with a generative network such that the network could reconstruct the original data. The generalization aspect of the neural network enforces similar data points to be close in the embedding space. Since our method includes the generative part, it allows visualizations that are not possible with neighborhood embedding methods such as TSNE. Compared to parametric methods such as VAE our method is non-parametric and relaxes the need to optimize the encoder part, allowing us to obtain better optimizations.

## 1    Introduction

Visualization is an essential tool for assessing the quality of feature representations, understanding groups and sub-groups in data, and comparing difficulties of decision boundaries across different classes. On the other hand, high-dimensional data visualization is challenging due to the curse of dimensionality. Thus, this is still an ongoing research direction in the machine learning community.

TSNE and UMAP are the common choices for high dimensional data visualization in recent machine learning literature (McInnes et al., 2018; Van der Maaten & Hinton, 2008). They create a neighborhood graph of the high-dimensional data and reconstruct these graph relationships in lower-dimensional space by gradient-based optimization.

Similar to neighborhood graph approaches, complete parametric methods such as Variational Auto Encoder (VAE) could provide visualizations as well if the bottleneck dimensions are restricted to two dimensions (Kingma & Welling, 2014). The visualization layer is completely controlled by the encoder portion of the network, thereby preventing any optimization for visualization independent for individual data points.

In this paper, we propose a method to create a visualization while obtaining a generative model of the data without an encoder. To achieve this, we relaxed the encoder part of Variational Auto Encoder and let it be learned as embeddings. We name this class of models as Generative Neural Embeddings (GNE). The optimization objective maximizes the embedding locations and generator (decoder) network jointly. We compared GNE generated visualizations versus existing methods. Additionally, we demonstrated the ability to generate new samples from embeddings.

## 2    Related Work

Our work is related to Generative Adversarial Networks (Goodfellow et al., 2014). In these methods, the networks are optimized in pairs as predictor and generator jointly. The generator part tries to map random noise into realistic data such that the discriminator would fail to discriminate between synthetic and real data. In our method, however, the generator tries to reconstruct the original images themselves. Also, embeddings are optimized instead of sampling from a random variable.

Word2vec (Mikolov et al., 2013b;a) uses a similar idea that models the generation of textual data by hierarchical softmax given the word embeddings. In our case, we have used a deep residual network for image generation and gave dummy ids to each image data point. Additionally, low-dimensional embeddings are chosen for visualization purposes.

DeepDream also optimizes the input image to maximize the activation of selected neuron (Mordvintsev et al., 2015). In our case, we are optimizing input embeddings of the generative model while allowing optimization on the generative model as well.

## 3 METHOD

Our method could be described as an embedding layer and a generative model on this embedding layer. Embedding layer $E$ provides the lookup table of embeddings for data point id $i$. Generative model $G$ generates the output from given embedding. Thus the generation process for data $i$ becomes $G(E[i])$. The objective of the whole network is to try to minimize the loss between generated data versus the real data corresponding to id $i$. The purpose of the embedding is to make the inputs of the generator optimizable. Hence, the name generative network embeddings (GNE) comes from this connection. Since the number of parameters grows with the number of data points, this method is a non-parametric method.

We have selected resnet structure for the generative part of the model (He et al., 2016). In order to have enough number of hidden units, there is a dense expansion layer that increases the number of dimensions after the embeddings. Embeddings are selected to be in 2 dimensions that could be displayed in a plane. However, any number of dimensions is possible for other purposes. In our model, we have an additional Gaussian noise layer that regularizes the embedding space. A Keras implementation of the model is given in the listing.

```
lin = Input(1)
embed = Embedding(x.shape[0], 2)(lin)
flatten = Flatten()(embed)
noise = GaussianNoise(1)(flatten)
hidden = Dense(N_HIDDEN, activation='elu')(noise)
for l in range(NLAYERS):
    relu = Dense(N_HIDDEN, activation='relu')(hidden)
    linear = Dense(N_HIDDEN)(relu)
    hidden = add([hidden, linear])
dense = Dense(OUTPUT_SHAPE, activation='sigmoid')(hidden)
```

## 4 RESULTS AND DISCUSSION

We used the MNIST training dataset to test the proposed method after normalizing images on 0-1 scale. Embedding size becomes 60000x2 for this dataset. We used Adam optimizer for training with an initial learning rate of 1e-2 with a batch size of 1024. We selected 64 dimensions as the number of hidden dimensions for 4 layers of residual blocks. We used mean squared error as a loss function.

Figure 1 shows five visualizations of this data. In the scatter plots, each color represents a distinct digit class, and the coordinates are embedding values for the corresponding digit image. Actual digit images are placed at corresponding coordinates. Figure 1a shows TSNE visualization. Distinct clusters are created which could allow discovery of certain classes of images in the absence of labels (colors). However, this visualization alone is not suitable for efficient exploration of images. In order to achieve this, generator networks that map 2d input coordinates to generated data samples can be used as shown in Figures 1c and 1e. In these plots, the range of maximum and minimum values is divided by 32 equal grid points which sent as an input to generator decoders.

In Figure 1b and 1c correspond to GNE optimization. In Figure 1d and 1e we show an equivalent network with 2d bottleneck layer. One difference between these two methods is the relative proximity of the various classes that they achieve. For example in VAE optimization, the images of digit 1 are adjacent to those of digit 5 which is not the case for GNE optimization. That adjacency between 1 and 5 can be observed grid plot in Figure 1e. Another difference is the fact that GNE relaxes the

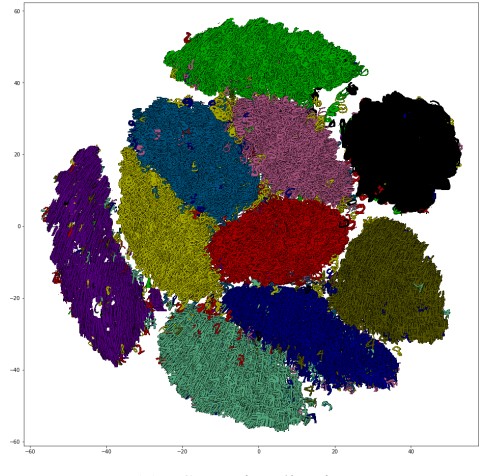

(a) TSNE visualization

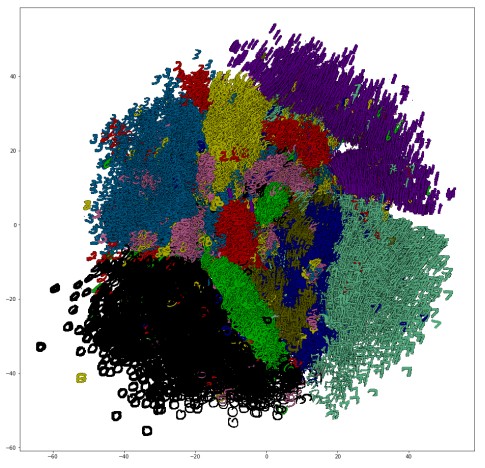

(b) GNE visualization

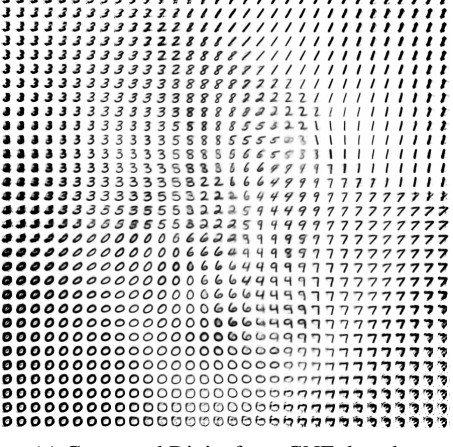

(c) Generated Digits from GNE decoder

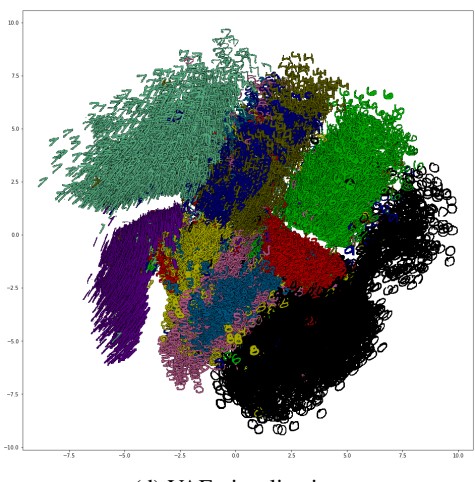

(d) VAE visualization

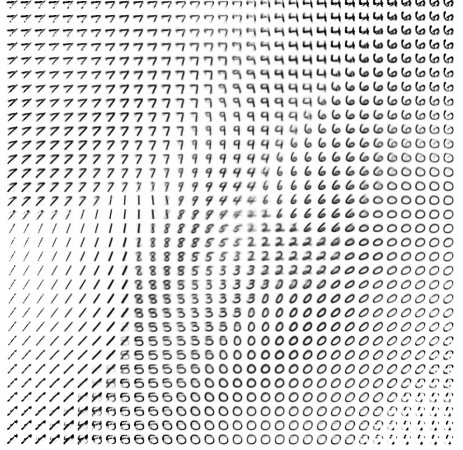

(e) Generated Digits from VAE decoder

need of optimizing encoder part which in our case allowed us to obtain lower loss function after training (GNE: 0.0288, VAE:0.0438 ).

## 5 FUTURE WORKS

We have demonstrated the utilization of embeddings to generate visualizations of data along with a generative model. There are additional potential aspects for improvement. We have listed them in this section.

First, the method could model each class using multi-modal distributions, and there is no explicit restriction factor that enforces the similar points to appear in a similar location in visualization space. On the other hand, adding pairwise similarity checks like in neighborhood graph models increases theoretical computational time.

Second, the optimization algorithm is gradient-based in this study. It does not have to be since the input space is very small. For example, even a global optimization with a simple grid search in 2D space could yield a better grouping of data points. However, this optimization diverges the optimization of the decoder part of the network from embeddings.

Third, GNE does not describe a direct way for getting embeddings for test data in the case of train-test split. One way to obtain them is by reusing embedding vectors for test cases and optimizing only the embeddings part. As expected, the optimization approach would be slower than standard feed-forward architectures.

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
