# OpenReview forum: "Deep Generative Neural Embeddings for High Dimensional Data Visualization"
_ICLR.cc/2022/Workshop/DGM4HSD — Submitted to ICLR 2022 DGM4HSD workshop_

### Official Review · Reviewer_BmMy · 2022-03-21
**Existing method, limited evaluation**

**Rating:** 3
**Confidence:** 5

**Review:**

The authors consider an auto-encoder where we parametrize the latent codes of training points _directly_, instead of parametrizing an encoder. The model is evaluated on data visualization, where we aim to visualize the structure in the data by plotting the two-dimensional latent codes.

Unfortunately, the proposed model is not novel, and has been explored in [1], where it is called an auto-_decoder_, and is used to learn a latent space of three-dimensional shapes. I assume the authors were not aware of this prior work.

The novelty of this paper is in using the method for data visualization, instead of generation/in-painting as in [1]. The MNIST experiment shows that the method indeed fits _reasonable_ two-dimensional codes. Unfortunately, the reader does not learn much beyond this. It is not clear if the learned codes are in any way more informative/useful than the codes learned by a standard auto-encoder, a VAE or TSNE/UMAP.

In summary, the proposed method already exists in literature, and the experimental aspect is too limited to constitute a significant contribution. I recommend rejecting the paper, but encourage the authors to continue studying the strengths/drawbacks of this method in the context of data visualization.

- [1] Park, J., Florence, P., Straub, J., Newcombe, R., & Lovegrove, S. (2019). _DeepSDF: Learning Continuous Signed Distance Functions for Shape Representation_. In The IEEE Conference on Computer Vision and Pattern Recognition (CVPR). https://openaccess.thecvf.com/content_CVPR_2019/html/Park_DeepSDF_Learning_Continuous_Signed_Distance_Functions_for_Shape_Representation_CVPR_2019_paper.html

---

### Official Review · Reviewer_5Mp2 · 2022-03-27

**Rating:** 3
**Confidence:** 4

**Review:**

This work presents a generative model for visualizing high-dim data. Overall the presentation is clear. However, I in person believe this is too premature in its current form, considering the following points:
- The generative nature of the model is not clear. The extact training objective seems to be mean squared error. In other words, this model only reconstructs the input in a point-wise manner, rather than learns the distribution of the input data.
- The results look weird for data visualization. As seen in Figure (b), some data belonging to the same class scatter around, while the embedding space learned by tSNE is much more compact.

---

### Official Review · Reviewer_84D2 · 2022-03-28
**Generative neural embedding**

**Rating:** 2
**Confidence:** 5

**Review:**

This submission is very badly written, it is not clear even on a basic level what is going on. The allusions to word2vec perhaps seem as if there is a 1-hot encoding of datapoints in the first layer which are embedded and then penalized to generate other datapoints, but which ones? neighbors? distal points? If it is just the point itself then this is no different than an autoencoder with a reconstruction loss. This has to be clarified.

The results of the MNIST embedding look very unconvincing as tSNE is showing better seperation. Also to compare visualizations I would suggest both looking at Moon et al. https://www.nature.com/articles/s41587-019-0336-3  for both the PHATE method as well as metrics such as ARI and DeMAP that are used for comparison.

---

### Decision · Program_Chairs · 2022-03-27

**Decision:**

Reject

**Comment:**

Although data visualization is an interesting topic, the AC agrees with both reviewers that the submission is not ready for acceptance at a workshop. We encourage the authors to take into account the reviewer's suggestions for a future submission.